# Fault Diagnosis of Mine Truck Hub Drive System Based on LMD Multi-Component Sample Entropy Fusion and LS-SVM

**Le Xu** [1,2]**, Wei Li** [1]**, Bo Zhang** [3,*]**, Yubin Zhu** [2] **and Chaonan Lang** [2]

1 School of Mechatronic Engineering, China University of Mining and Technology, Xuzhou 221116, China; lb20050019@cumt.edu.cn (L.X.); liwei_cmee@163.com (W.L.)
2 Engineering Training Center, Jiangsu Normal University, Xuzhou 221116, China; 6020170059@jsnu.edu.cn (Y.Z.); 6020180108@jsnu.edu.cn (C.L.)
3 School of Computer Science and Technology, China University of Mining and Technology, Xuzhou 221116, China
* Correspondence: zbcumt@163.com

**Abstract:** As the main transportation equipment in ore mining, the wheel drive system of mining trucks plays a crucial role in the transportation capacity of mining trucks. The internal components of the hub drive system are mainly composed of bearings, gears, etc. The vibration signals caused during operation are nonlinear and nonstationary complex signals, and there may be more than one factor that causes faults, which causes certain difficulties for the fault diagnosis of the hub drive system. A fault diagnosis method based on local mean decomposition (LMD) multi-component sample entropy fusion and LS-SVM is proposed to address this issue. Firstly, the LMD method is used to decompose the vibration signals in different states to obtain a finite number of PF components. Then, based on the typical correlation analysis method, the distribution characteristics and correlation coefficients of vibration signals in the frequency domain under different states are calculated, and effective PF multi-component sample entropy features are constructed. Finally, the LS-SVM multi-fault classifier is used to train and test the extracted multi-component sample entropy features to verify the effectiveness of the method. The experimental results show that, even in small-sample data, the LMD multi-component sample entropy fusion and LS-SVM method can accurately extract fault features of vibration signals and complete classification, achieving fault diagnosis of wheel drive systems.

**Keywords:** local mean decomposition (LMD); sample entropy; fault diagnosis; mine truck hub drive system





## 1. Introduction

Mining trucks are important equipment for mining production and transportation, undertaking the transportation of coal and 90% of iron ore open-pit mining in the world. Mining trucks often need to work continuously in harsh environments full of noise, floating dust, and turbulence [1,2]. The noise, vibration, high temperature, high humidity, and other factors existing in the working environment cause great harm to the health of these giant pieces of mining equipment. In order to ensure safe and efficient mining, it is necessary to carry out intelligent monitoring on the hub of a mining truck and use data information depth mining to achieve fault diagnosis [3,4]. The failures in the hub drive system of mining trucks mainly include bearing failures, gear failures, and other failures, among which the failure probability of bearing and gear failures is up to 90%. When the wheel hub drive system fails, the vibration signal caused by either the bearing or the gear is nonlinear and nonstationary, so it is necessary to use corresponding methods to extract the characteristics of the vibration signal. Common fault feature extraction methods include time domain analysis, wavelet transform, and empirical mode decomposition. However, these methods

have their own limitations in signal processing, especially in the adaptive effect of wheel drive vibration signal analysis with large data amplitude.

## 2. Literature Analysis

Vibration signal analysis is a commonly used monitoring technology in mechanical systems. It contains a great deal of fault information and has a wide range of applications in the field of fault diagnosis of rotating machinery such as bearings and gears. Because of the nonlinear and nonstationary characteristics of the vibration signals of the bearings and gears in the hub drive system, the traditional time domain analysis methods cannot meet the needs of the fault diagnosis of the hub drive system. The time–frequency analysis methods represented by empirical mode decomposition (EMD) [5], empirical wavelet transform (EWT) [6], and local mean decomposition (LMD) [7] have been widely used in the field of bearing and gear fault diagnosis. Debiao Meng [8] proposed a fault analysis method of wind power rolling bearing based on EMD feature extraction and achieved ideal results. To solve the problem of feature extraction of a weak fault of a rolling bearing, Yin Chen [9] combined empirical mode decomposition and adaptive threshold denoising (ATD) to automatically extract the inherent noise hidden in the original signal, providing support for weak fault features of rolling bearings. For the gearbox running under nonstationary working conditions, Ridha Ziani [10] used EMD, TKEO, and Shock Detector to realize the damage detection of a helical gearbox running under variable load and speed. Although the EMD method can extract the fault features of bearing and gear vibration signals, the EMD method has some shortcomings [11], such as mode aliasing and endpoint effect, which affects the effect of fault feature extraction. Empirical wavelet transform (EWT) [12] is a combination of wavelet transform and EMD. Chegini [13] used an EWT denoising method to carry out fault vibration of rolling bearings. The results show that the EWT denoising method is superior to EMD denoising technology. Hu Mantang [14] optimized EWT and solved the problem that the failure samples of early bearing vibration signals were overwhelmed by normal samples. Although EWT is better than EMD to some extent, EWT vibration signal decomposition is affected by the adaptive and robust boundary of EWT, and the filtered signal still contains residual noise, which covers the fault feature signal and affects the feature extraction effect [15,16]. Local mean decomposition (LMD) is a signal analysis method proposed by Smith S [17] for the first time. It can adaptively decompose complex multi-component signals into the sum of several product functions (PF) [18]. This method has strong adaptability to nonlinear and nonstationary signal analysis [19]. Minghong Han [20] used the combination method of LMD and multi-scale symbolic dynamic information entropy (MSDE) to diagnose and analyze the fault type and degree of rolling bearing and achieved good results. Song Enzhe [21] improved LMD, combined with composite multi-scale weighted permutation entropy (CMWPE) and support vector machine (SVM), and accurately distinguished various fault types of rolling bearings under the same fault degree, with more reliable diagnosis results. Compared with the EMD method, the LMD method can effectively suppress the end effect and solve the problems of under-envelope and over-envelope [22,23]. Compared with the EWT method [24], the LMD method has fewer iterations in the signal analysis process, avoiding the generation of multiple false components in the decomposition process. However, there is a slight error between the local mean function and the envelope estimation function in the LMD decomposition process and the actual situation, which will also lead to mode confusion, reducing the accuracy of LMD diagnosis [25,26]. Entropy [27,28] is a quantity used to describe the uncertainty in a data distribution in a system, which can measure the degree of data distribution disorder and effectively represent the distribution of internal quantities of the system. Sample entropy (SE) is a method proposed by Richman [29] to measure the complexity of signal sequences based on approximate entropy. When the signal sequence is complex, its own signal similarity is low, and the corresponding sample entropy is large. When the signal sequence is not complex, its own signal similarity is high, and the corresponding sample entropy is small [30,31]. Li Xuguang [32] extracted

fault features from the processed rolling bearing signals through MEEMD sample entropy and realized rolling bearing fault diagnosis. Zhang Decai [33] extracted gearbox fault features using Euclidean matrix sample entropy and realized gearbox composite fault diagnosis with a one-dimensional convolutional neural network. After a failure of the wheel hub drive system, the vibration signal caused by its operation will change, and the time series represented by the vibration signal will change with the failure. Therefore, the fault characteristics of the wheel hub drive system vibration signal can be extracted by using the sample entropy feature. In practice, the effect of fault feature extraction using single-sample entropy of a vibration signal is not ideal, and there is still a certain gap in the accuracy of diagnosis. Therefore, this paper proposes a multi-scale sample entropy method, which combines the PF component after LMD processing and typical correlation analysis to achieve fault feature extraction.

As it is difficult to accurately diagnose a fault of the nonlinear and nonstationary vibration signal of the wheel hub drive system, the LMD method is used to analyze the vibration signal of the wheel hub drive system, and several PF components are obtained. Through canonical correlation analysis, the sub-signals with high correlation coefficients are analyzed and the sample entropy value is calculated and the multi-component sample entropy feature vector is constructed to realize the fault feature extraction of the wheel hub drive system. At the same time, in response to the small amount of vibration signal acquisition in the hub drive system, this paper uses LS-SVM to train and test the fault features extracted from the vibration signal of the hub drive system using LMD multi-component sample entropy fusion, achieving fault recognition and classification of the hub drive system. The contributions of this article are as follows:

(1) A feature extraction method based on LMD multi-component sample entropy fusion is proposed. Aiming at the problems of mode confusion and poor accuracy in LMD decomposition, canonical correlation analysis (CCA) [34,35] is used to discriminate the true and false components of the decomposed PF, and then the multi-component sample entropy fusion sample entropy feature is constructed.

(2) Combining the vibration signal characteristics of the wheel drive system, the LMD multi-component sample entropy fusion feature is introduced into the fault diagnosis of nonstationary and nonlinear vibration signals in the wheel drive system, better characterizing the fault feature information.

(3) In response to the difficulty in obtaining vibration signals from the wheel drive system and the small number of samples, LS-SVM is proposed to classify fault features using LMD multi-component sample entropy fusion features extracted from the vibration signals of the wheel drive system, which improves the accuracy of the algorithm.

(4) The effectiveness of this method has been verified through experiments.

## 3. Theory

### 3.1. Local Mean Decomposition

LMD is a complex signal adaptive decomposition process, which adaptively decomposes nonlinear and nonstationary signals into several PF components. Each PF component is obtained by multiplying the corresponding envelope signal and pure frequency modulation signal. When LMD algorithm is used for signal decomposition to obtain K PF components, the corresponding adaptive decomposition model is as follows [17]:

Mark all extreme points $n_i$ ($i$ = 1, 2, 3...) of signal $x(t)$, and calculate the mean value $m_i$ between adjacent extreme points $n_i$ and $n_{i+1}$ and their envelope estimation value $a_i$.

$$m_i = \frac{n_i + n_{i+1}}{2} \tag{1}$$

$$a_i = \frac{|n_i - n_{i+1}|}{2} \tag{2}$$

Connect all the calculated mean values $m_i$ and envelope estimation values $a_i$ in turn and use the moving average method to process them respectively to obtain the local mean function $m_{11}(t)$ and envelope estimation function $a_{11}(t)$.

The local mean function $m_{11}(t)$ is separated from the original signal $x(t)$ to obtain

$$h_{11}(t) = x(t) - m_{11}(t) \tag{3}$$

The envelope function $a_{11}(t)$ is used to demodulate the obtained $h_{11}(t)$ to obtain the frequency modulation signal $s_{11}(t)$.

$$s_{11}(t) = \frac{h_{11}(t)}{a_{11}(t)} \tag{4}$$

In an ideal state, $s_{11}(t)$ is a pure frequency modulation signal, so its corresponding envelope estimation function $a_{12}(t) = 1$. If the envelope estimation function $a_{12}(t) \neq 1$, treat $s_{11}(t)$ as the original signal and repeat the above iterative steps until the pure frequency modulation signal $s_{1n}(t)$ is obtained, and then $-1 \leq s_{1n}(t) \leq 1$ can be satisfied, and the corresponding envelope estimation function $a_{1(n+1)}(t) = 1$. The specific steps are as follows:

$$\begin{cases} h_{11}(t) = x(t) - m_{11}(t) \\ h_{12}(t) = s_{11}(t) - m_{12}(t) \\ \cdots \\ h_{1n}(t) = s_{1(n-1)}(t) - m_{1n}(t) \end{cases} \tag{5}$$

$$\begin{cases} s_{11}(t) = \frac{h_{11}(t)}{a_{11}(t)} \\ s_{12}(t) = \frac{h_{12}(t)}{a_{12}(t)} \\ \cdots \\ s_{1n}(t) = \frac{h_{1n}(t)}{a_{1n}(t)} \end{cases} \tag{6}$$

Theoretically, $a_{1(n+1)}(t) = 1$ is the ideal state for obtaining pure frequency modulation signal $s_{1n}(t)$. In practice, in order to reduce the number of iterations and improve the calculation efficiency, a small deviation $\Delta$ ($\Delta > 0$) is introduced without changing the decomposition results. When $1 - \Delta \leq a_{1(n+1)}(t) \leq 1 + \Delta$, $s_{1n}(t)$ is considered to be a relatively ideal pure frequency modulation signal. With reference to the literature and a large amount of experimental data, deviation $\Delta$ is the most appropriate value in the range of [0.001, 0.1]. In this paper, under the condition that the iteration results are correct and meet the needs of feature extraction, deviation $\Delta$ is taken as 0.05. Then, the above iteration termination condition is as follows:

$$0.95 \leq a_{1(n+1)}(t) \leq 1.05 \tag{7}$$

The envelope signal $a_1(t)$ can be obtained by multiplying all envelope functions obtained before the end of iteration.

$$a_1(t) = a_{11}(t)a_{12}(t) \cdots a_{1n}(t) = \prod_{i=1}^{n} a_{1i}(t) \tag{8}$$

The first PF component decomposed by $x(t)$ can be obtained by multiplying $a_1(t)$ and $s_{1n}(t)$.

$$PF_1(t) = a_1(t)s_{1n}(t) \tag{9}$$

$PF_1(t)$ is separated from $x(t)$, and the remaining signals are recorded as $u_1(t)$. Repeat the above steps with signal $u_1(t)$ as a new signal for $k$ times until $u_k(t)$ is a monotone function, and then the extreme point $u_k(t) \leq 1$.

$$\begin{cases} u_1(t) = x(t) - PF_1(t) \\ u_2(t) = u_1(t) - PF_2(t) \\ \cdots \\ u_k(t) = u_{k-1}(t) - PF_k(t) \end{cases} \tag{10}$$

After completing the above steps, the signal $x(t)$ will be decomposed into $k$ PF components and a residual value $u_k(t)$, which is shown as follows:

$$x(t) = \sum_{i=1}^{k} PF_i(t) + u_k(t) \tag{11}$$

### 3.2. Sample Entropy

Entropy is a quantity used to describe the uncertainty in the data distribution in a system, which can measure the degree of data distribution disorder and effectively represent the distribution of internal quantities of the system. Sample entropy is a method to measure the complexity of signal sequence. When the signal sequence is complex, its own signal similarity is low, and the corresponding sample entropy is large. When the signal sequence is not complex, its own signal similarity is high, and the corresponding sample entropy is small.

For the known time series $x(N) = \{x(1), x(2),\ldots, x(n)\}$, the sample entropy calculation method is as follows [29]:

The $m$-dimensional matrix sequence $X_m(i)$ is constructed according to the serial number.

$$X_m(i) = [x(i), x(i+1), \cdots, x(i+m-1)] \quad i = 1, 2, \cdots, n-m+1 \tag{12}$$

Define $d[X_m(i), X_m(j)]$ as the maximum difference between the two vectors $X_m(i)$ and $X_m(j)$.

$$d[X_m(i), X_m(j)] = \max_{k=0 \rightarrow m-1} |x(i+k) - x(j+k)| \tag{13}$$

For $X_m(i)$, the number of $d[X_m(i), X_m(j)]$ less than $r$ is recorded as $B_i$, where $r$ is the similarity threshold. Record the ratio of $B_i$ to the number of vectors as $B_i^m(r)$.

$$B_i^m(r) = \frac{B_i}{n-m+1} \quad 1 \leq i \leq n-m \tag{14}$$

$$B^m(r) = \frac{1}{n-m+1} \sum_{i=1}^{n-m+1} B_i^m(r) \tag{15}$$

Find the mean $B^m(r)$ of $B_i^m(r)$.

By increasing the matrix to $m + 1$ dimension and repeating steps (1) to (4) above, $B^{m+1}(r)$ can be obtained.

In actual calculation, $N$ is taken as a limited quantity, and the corresponding sample entropy is calculated as follows:

$$SampEn(m, \; r, \; N) = -\ln \frac{B^{m+1}(r)}{B^m(r)} \tag{16}$$

The sample entropy calculation results have a great relationship with the values of $m$ and $r$ in the above steps. In this paper, according to the research results and research objects, when $m = 2$, $r = 0.15 \, Std$ is selected, the sample entropy value calculated is the most appropriate (Std is the standard deviation of the signal sequence).

### 3.3. Canonical Correlation Analysis

The CCA algorithm aims to find the projection vector of two data volumes, which can maximize the correlation between the two data. CCA algorithm is described as follows [35]:

Given two sets of random datasets $X = [x_1, x_2, \cdots, x_N]$ and $Y = [y_1, y_2, \cdots, y_N] \in R^{d_y \times N}$, find two projection vectors $\omega_x \in R^{d_x}$ and $\omega_y \in R^{d_y}$ to maximize the correlation coefficient between $\omega_x^T X$ and $\omega_y^T Y$.

$$\max_{\omega_x, \omega_y} \frac{\omega_x^T C_{xy} \omega_y}{\sqrt{\left(\omega_x^T C_{xx} \omega_x\right)\left(\omega_y^T C_{yy} \omega_y\right)}} \tag{17}$$

In the above equation

$$\begin{cases} C_{xx} = \frac{1}{N-1}\left(XX^T\right) \\ C_{xy} = \frac{1}{N-1}\left(XY^T\right) \\ C_{yy} = \frac{1}{N-1}\left(YY^T\right) \end{cases} \tag{18}$$

Because $\omega_x$ and $\omega_y$ are scale-invariant, the above equation can be written as follows:

$$\max_{\omega_x,\omega_y} \omega_x^T C_{xy} \omega_y$$
$$\text{s.t. } \omega_x^T C_{xx} \omega_x = \omega_y^T C_{yy} \omega_y = 1 \tag{19}$$

### 3.4. LMD Multi-Component Sample Entropy Fusion

The frequency components of the vibration signals caused by the gears in the drive system of the mine truck hub are different when they operate under normal and fault conditions. Moreover, under different faults, the arrangement and distribution complexity of frequency components will also change. In order to analyze the change in signal sequence of vibration signals in different frequency domains and quantitatively present the degree of distribution disorder in different frequency domains, LMD is used to decompose the original signal, and typical correlation analysis is conducted between the decomposed PF component and the original signal. The false components with small correlation are discarded and the components with large correlation with the original signal are taken as the analysis object. The sample entropy of signal sequences in different frequency domains is calculated respectively to form multi-component sample entropy fusion vector, which is used as the feature vector for fault feature extraction and classification. The specific methods are as follows:

The collected vibration signal is decomposed by LMD to obtain $k$ PF components and residual value $u_k(t)$. The correlation between $k$ PF components and the original signal is analyzed by CCA method. The PF component with large correlation coefficient (the real component in the effective frequency domain) is taken as the analysis object, and then the sample entropy of the effective PF component is calculated. Since the original signal contains false invalid signals, a single PF component can only characterize the fault characteristics in the corresponding frequency domain. Therefore, the sample entropy corresponding to the effective PF component of LMD decomposition is used to construct a feature vector, and this vector is used as the fault feature for analysis. LMD multi-component sample entropy fusion feature extraction model is shown in Figure 1.

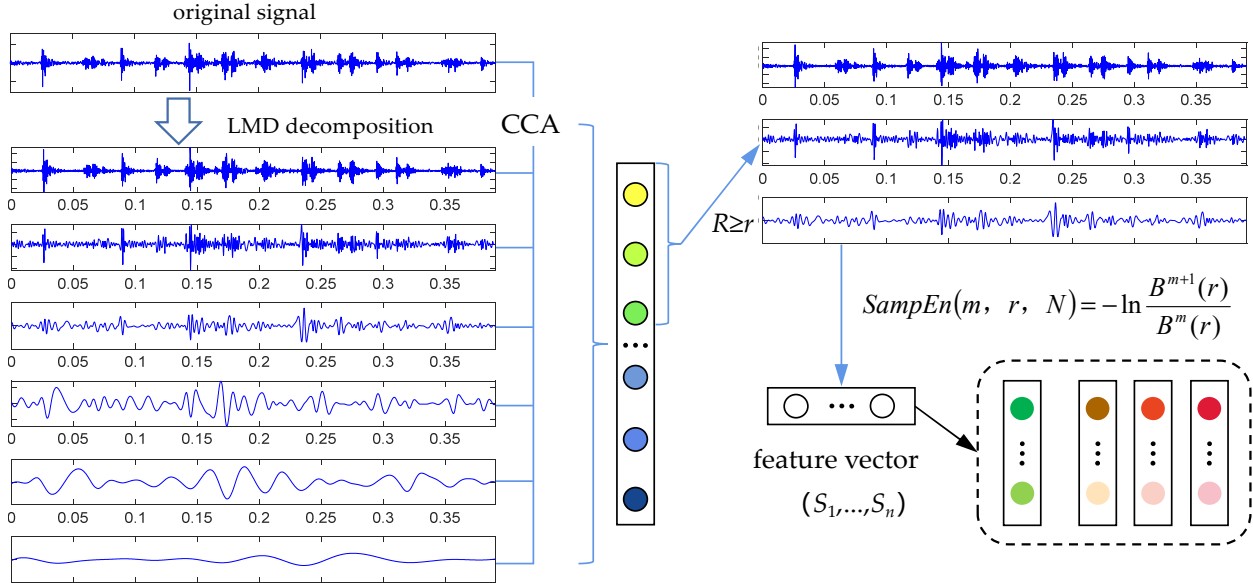

**Figure 1.** LMD multi-component sample entropy fusion feature extraction model.

*3.5. LS-SVM*

Support vector machine (SVM) is a kernel-based machine learning method that does not require a large number of training and testing samples. It has high accuracy in fault classification and is very suitable for small-sample data analysis and processing. Least squares support vector machine (LS-SVM) is a simplified version of SVM, which transforms quadratic programming in SVM into solving a system of linear equations, reducing the complexity of the solving process, shortening training time, and improving recognition accuracy. LS-SVM can be expressed as follows [19]:

For dataset $N = \{(x_i, y_i) \ i = 1, 2, \cdots, n\}$, the linear regression function is represented as follows:

$$y(x_i) = \omega^T \varphi(x_i) + b \tag{20}$$

The $b$ is bias, $\omega$ is the weight coefficient vector, and $\varphi(x_i)$ is a nonlinear function.

The optimization function corresponding to LS-SVM is as follows:

$$min\left(\frac{1}{2}\|\omega\|^2 + \frac{1}{2}\gamma \sum_{i=1}^{n} \varepsilon_i^2\right) \tag{21}$$

The constraints are as follows:

$$y_i = \omega^T \varphi(x_i) + b + \varepsilon_i \quad (i = 1, 2, \cdots, n) \tag{22}$$

The $\varepsilon_i$ is the error variable and $\gamma$ is the penalty factor.

The optimal solution can be calculated using the dual method, and the Lagrangian function can be introduced based on the dual method as follows:

$$L = \frac{1}{2}\|\omega\|^2 + \frac{1}{2}\gamma \sum_{i=1}^{n} \varepsilon_i^2 - \sum_{i=1}^{n} \alpha_i \left(\omega^T \varphi(x_i) + b + \varepsilon_i - y_i\right) \tag{23}$$

The $\alpha_i$ is a Lagrange multiplier.

According to the Karush–Kuhn–Tucker condition, take the partial derivatives of $\omega$, $b$, $\varepsilon_i$, and $\alpha_i$ in sequence and make them equal to 0.

$$\begin{cases} \frac{\partial L}{\partial \omega} = \omega - \sum_{i=1}^{n} \alpha_i \varphi(x_i) = 0 \\ \frac{\partial L}{\partial b} = \sum_{i=1}^{n} \alpha_i = 0 \\ \frac{\partial L}{\partial \varepsilon_i} = \alpha_i - \gamma \varepsilon_i = 0 \\ \frac{\partial L}{\partial \alpha_i} = y_i - \left(\omega^T \varphi(x_i) + b + \varepsilon_i\right) = 0 \end{cases} \tag{24}$$

Eliminate $\omega$ and $\varepsilon_i$, and further process them according to Mercer's theorem. The kernel function is represented as follows:

$$K(x_i, x_j) = \varphi(x_i)^T \varphi(x_j) \quad i, j = 1, 2, \cdots, n \tag{25}$$

The LS-SVM prediction function is represented as

$$y(x) = \sum_{i=1}^{n} \alpha_i K(x_i, x_j) + b \tag{26}$$

The commonly used kernel functions include linear kernel function, polynomial kernel function, and RBF kernel function.

## 4. Experimental Analysis

*4.1. Data Collection*

In order to simulate the fault characteristics of the internal rotating parts of the mine truck hub drive system and analyze the extraction and diagnosis effect of the fault characteristics based on the LMD multi-component sample entropy and fusion canonical correlation

analysis method, the gearbox vibration signal acquisition is selected in the rotating machinery fault simulation test bed, and its experimental device is shown in Figure 2.

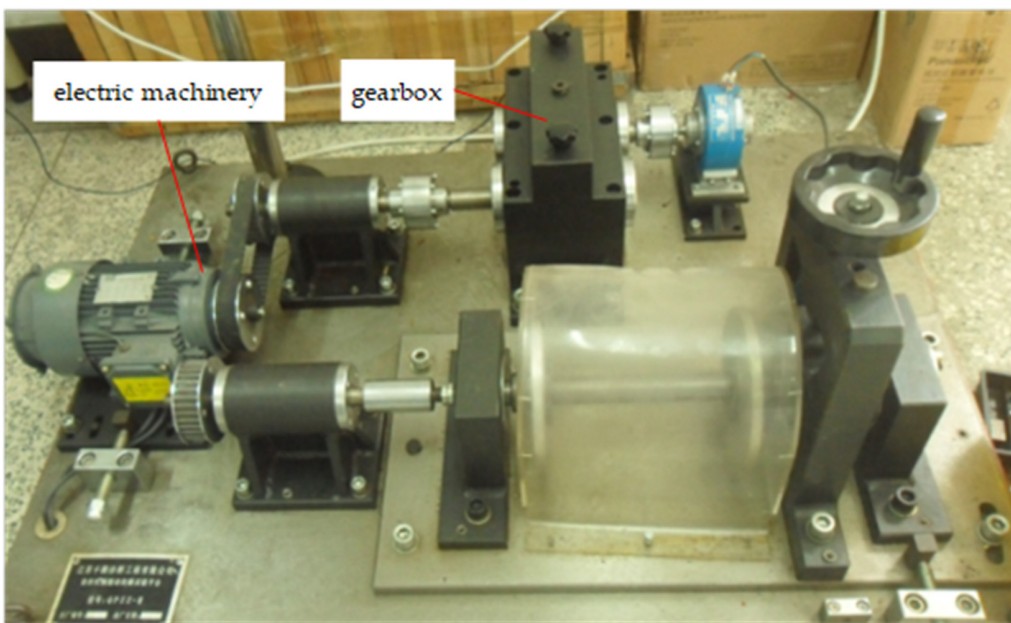

**Figure 2.** Rotating machinery fault simulation test bench.

Install one acceleration sensor in the horizontal and vertical directions of the gearbox cover and use ADA16-8/2 (LPCI) acquisition card to collect data. In the experiment, in order to simulate faults in the mining truck hub drive system, in addition to collecting normal gearbox vibration data, fault data were also collected by replacing different faulty large and small gears, including three types of faults: small gear breakage, big gear wear, small gear breakage + big gear wear. All four types of gears are standard spur gears, made of S45C material, with a modulus of 2. The number of teeth for the large gear is 75, and the number of teeth for the small gear is 55. The actual experimental gear is shown in Figure 3. The experimental process is as follows:

Step 1: After installing the faulty gear on the experimental platform, start the testing machine and check the following conditions: whether all bolts and screws are loose, whether the insulation resistance of the motor is greater than 20 M $\Omega$, whether there are any errors in the wiring of each part, etc. After confirming that there are no errors, stop the machine.

Step 2: Install the piezoelectric accelerometer on the gearbox base, and then connect the two signal wires to the signal processor.

Step 3: Open the data acquisition software and set the sampling frequency to 5120 Hz and the number of acquisition points to 2000. Start and run the machine. When the operation is stable, click "Sampling" to obtain the vibration signal and save it. In order to meet the requirements of small-sample data analysis (with no more than 30 samples in the same group), 20 sets of vibration data were collected to obtain normal gear vibration data.

Step 4: Open the gearbox housing and replace the normal small gear with a broken tooth small gear. In the same environment, repeat steps 1–3 to obtain 20 sets of broken tooth fault vibration data.

Step 5: Open the gearbox housing and replace the normal large gear with a worn large gear. In the same environment, repeat steps 1–3 to obtain 20 sets of broken teeth and wear fault vibration data.

Step 6: Open the gearbox housing and replace the broken tooth small gear with a normal small gear. In the same environment, repeat steps 1–3 to obtain 20 sets of wear fault vibration data.

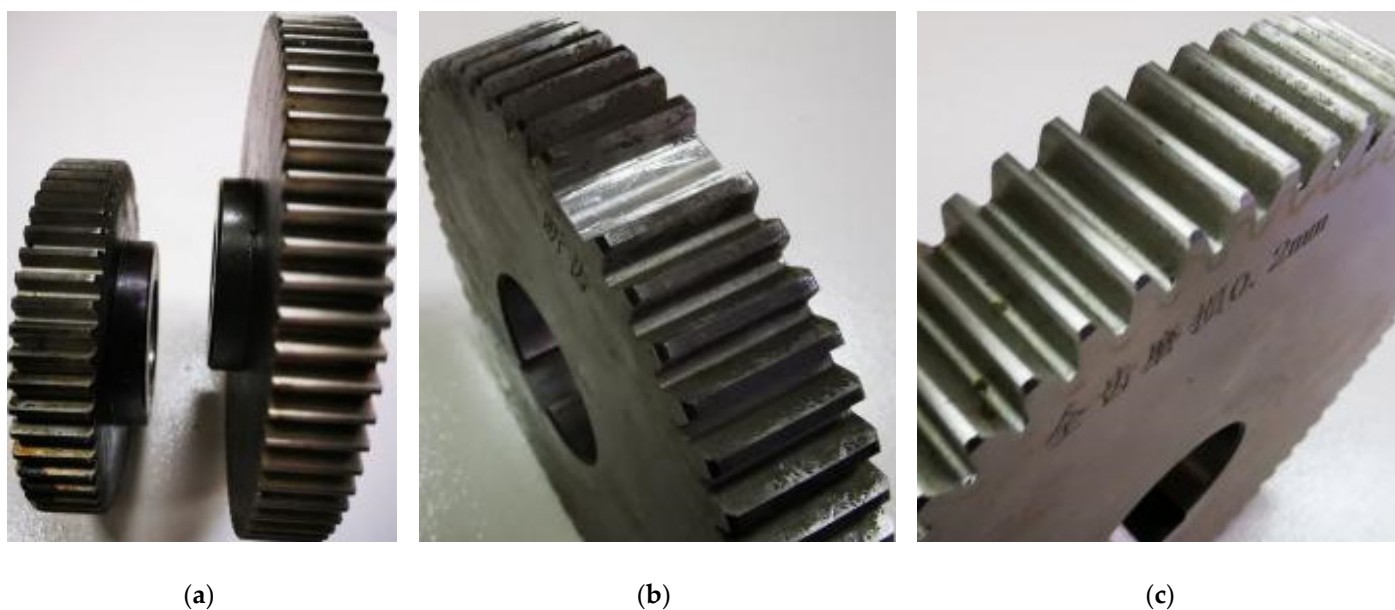

(**a**)                                          (**b**)                                          (**c**)

**Figure 3.** Experimental gear. (**a**) Normal; (**b**) broken teeth; (**c**) wear.

### 4.2. Vibration Signal Analysis

Follow the above steps to perform LMD decomposition on each group of gearbox vibration signals collected in sequence and obtain several PF components and one residual after decomposition. Figure 4 takes a set of wear faults as an example, where the original vibration signal is decomposed by LMD to obtain five PF components and one residual component. From the decomposition results, it can be seen that the various PF components after LMD decomposition separate the original signal in order of resolution from high to low. The residual $u_5(t)$ fluctuates about three times and the amplitude is very weak and can be regarded as a monotonic function when a small deviation of 0.05 $\Delta$ is added.

In order to select the PF component that can reflect the main vibration characteristic information from the components decomposed by LMD, canonical correlation analysis is used to analyze each PF component and the original signal. The CCA coefficient value that can reflect the real vibration characteristic component of the original signal is large, and the CCA coefficient value that cannot truly reflect the false, noise, and other components of the original signal vibration is small. Table 1 shows the canonical correlation coefficient values between the original vibration signals of the gearbox in different states under the same environment and the PF components after LMD decomposition.

From Table 1, it can be seen that the first three PF components of the original signal decomposed by LMD under different states have relatively high correlation with the original vibration signal, while the correlation coefficients of the last two PF components and residual quantities with the original signal are relatively small (not exceeding 0.05). By reviewing research materials and data analysis results, the first three PF components are considered as true and usable components and used as analysis elements for fault feature extraction.

**Table 1.** Correlation coefficient between each PF component and the original signal.

| Gear Type | Canonical Correlation | | | | | |
|---|---|---|---|---|---|---|
| | $PF_1$ | $PF_2$ | $PF_3$ | $PF_4$ | $PF_5$ | $u_5(t)$ |
| normal | 0.685 | 0.593 | 0.186 | 0.019 | 0.00023 | 0.00011 |
| broken teeth | 0.792 | 0.526 | 0.238 | 0.021 | 0.00017 | 0.00006 |
| wear | 0.612 | 0.624 | 0.195 | 0.012 | 0.00030 | 0.00017 |
| broken teeth + wear | 0.801 | 0.496 | 0.156 | 0.024 | 0.00013 | 0.00014 |

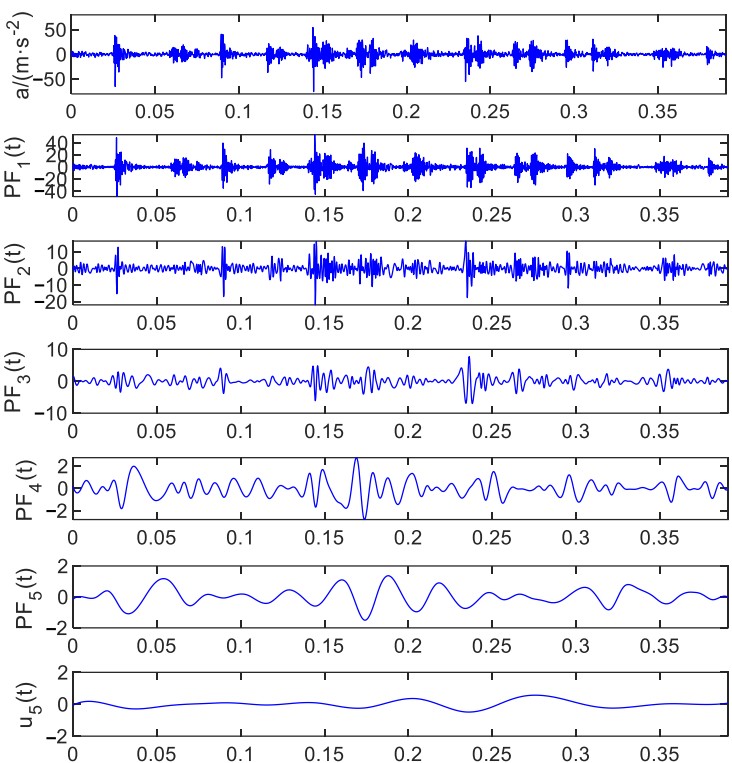

**Figure 4.** Wear state original vibration signal and LMD decomposition result.

### 4.3. Fault Feature Extraction

Select the top 10 sets of vibration signals from four different states as LS-SVM training samples. Figure 5 shows the entropy distribution of 40 sets of original vibration signal samples for the four states, and Figure 6 shows the entropy distribution of the first three PF components corresponding to the vibration signal after LMD decomposition. From the two graphs, it can be seen that, although the sample entropy distribution of the original vibration signal and the first three PF components can be maintained within a certain interval range, there is a jumping phenomenon; that is, the sample entropy in different states intersects and overlaps between intervals. However, from the distribution of LMD multi-component sample entropy fusion corresponding to the 40 sets of vibration signals in Figure 7, it can be seen that, although the entropy values of each PF component sample cross and jump, after fusing the sample entropy of the first three PF components into feature vectors, they will be concentrated in a certain spatial range, representing a relatively significant distribution pattern.

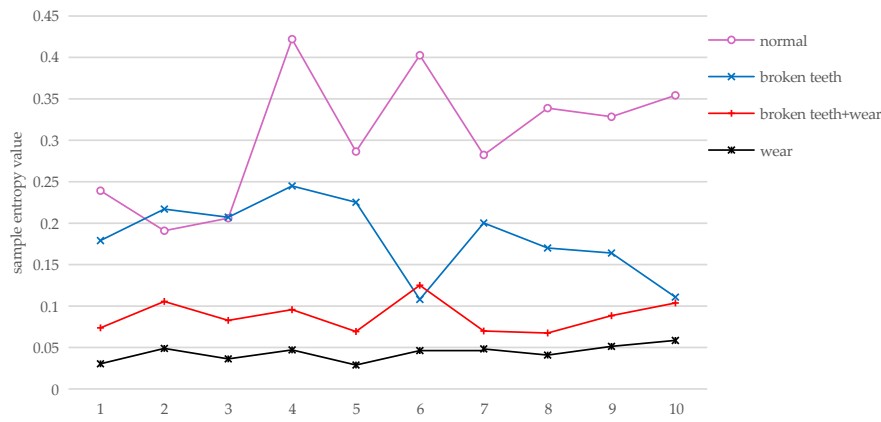

**Figure 5.** Sample entropy distribution of original vibration signal of training sample.

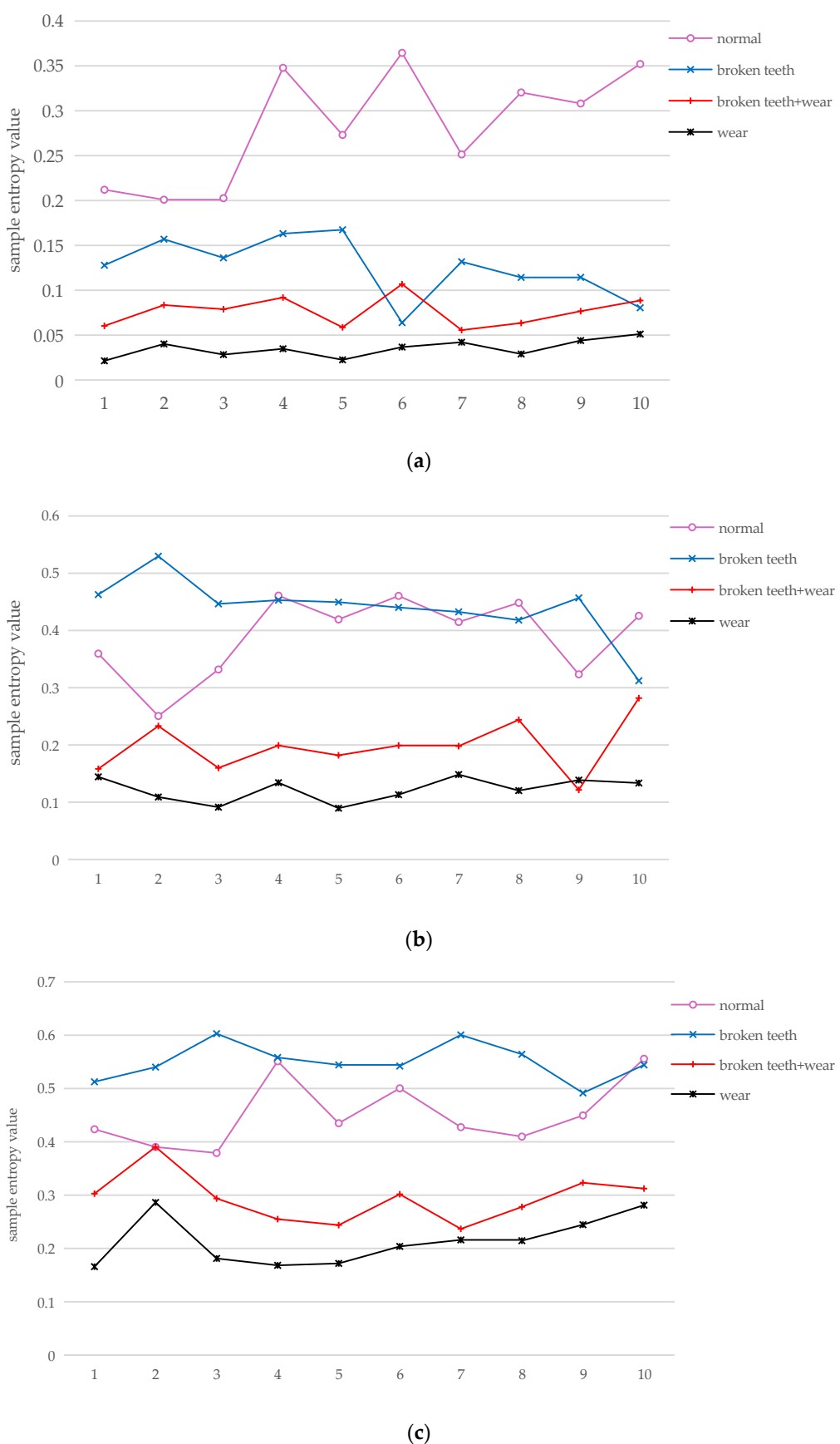

**Figure 6.** Entropy distribution of training samples PF1–PF3. (**a**) PF1; (**b**) PF2; (**c**) PF3.

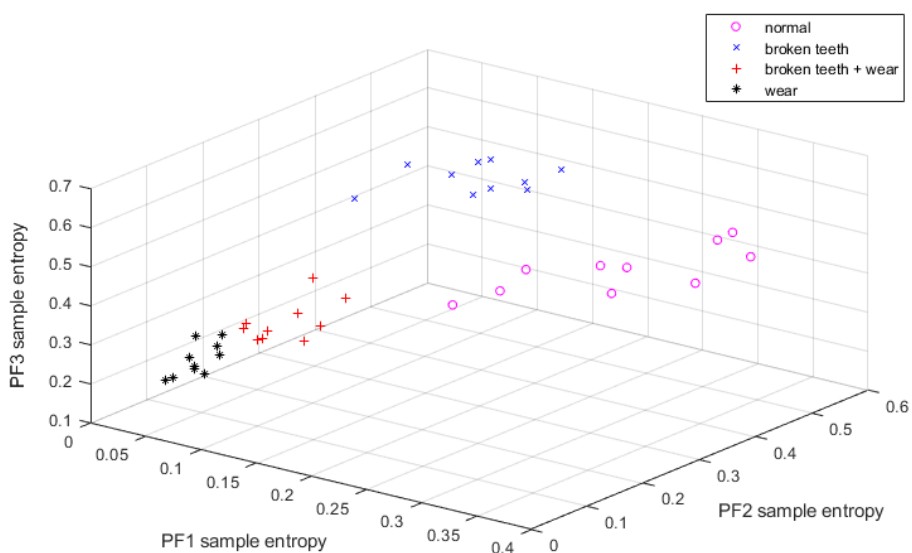

**Figure 7.** Training sample LMD multi-component sample entropy fusion distribution.

*4.4. Classification of Fault States*

Input the LMD multi-component sample entropy of the first 10 sets of vibration signals in four different states into the LS-SVM classifier for training. Input the last 10 sets of vibration signals in each state as test samples into the trained LS-SVM classifier for recognition and classification. Train and classify the LMD multi-component sample entropy fusion features of gearbox vibration signals in the above four states based on linear, polynomial, and RBF kernel functions, respectively. The training and classification results are shown in Table 2.

**Table 2.** Diagnosis effects of multi-component sample entropy fusion and LS-SVM based on LMD.

| Type | | Normal | Broken Teeth | Wear | Broken Teeth + Wear |
|---|---|---|---|---|---|
| linear kernel function | training time | 0.321 s | 0.332 s | 0.340 s | 0.343 s |
| | precision | 90% | 70% | 80% | 80% |
| Polynomial kernel function | training time | 0.364 s | 0.373 s | 0.387 s | 0.382 s |
| | precision | 90% | 80% | 80% | 90% |
| RBF kernel function | training time | 0.431 s | 0.457 s | 0.463 s | 0.475 s |
| | precision | 100% | 100% | 100% | 100% |

From the diagnostic results in Table 2, it can be seen that the training time based on the three kernel functions is relatively close, with the shortest based on linear kernel functions and the longest based on RBF kernel functions. However, in terms of recognition and classification accuracy, the RBF kernel function is higher than the other two, and the recognition accuracy for all four types of gearbox states reaches 100%. The main reason for the above classification results is that linear kernel function is the simplest kernel function, which can map data from low-dimensional space to high-dimensional space, but its monetization power is limited and only applicable to linearly separable data. Polynomial kernel functions can map data to higher dimensional spaces, but their expressive power is also limited and only applicable to some simple nonlinear problems. The RBF kernel function can map data to an infinite dimensional space and has strong expressive power, making it suitable for complex nonlinear problems. However, its training time will be slightly longer than the first two. The data collected in this experiment are nonlinear and nonstationary vibration signals, so the effect of using the RBF kernel function will be more significant. Due to the fact that the faulty gears used in the experiment were artificially processed, the fault characteristics caused by vibration are more obvious, and the experimental environment is relatively stable. Therefore, the collected data are idealized, and

the test results are also idealized. In real environments, there may be certain differences in the results due to various factors, such as fault type, degree of damage, and usage environment.

## 5. Conclusions

A fault diagnosis method based on LMD multi-component sample entropy fusion and LS-SVM is proposed to address the difficulties caused by nonlinear and nonstationary vibration signals inside the mining truck wheel drive system in fault diagnosis. This method can provide reference for the fault diagnosis of rolling bearings and gears inside the wheel drive system and is of great significance for improving the efficiency of mining trucks and reducing operation and maintenance costs. The main conclusions are as follows:

(1) The proposed LMD multi-component sample entropy fusion can effectively extract fault diagnosis features within the wheel drive system, which has significant advantages compared to traditional methods.

(2) Introducing LS-SVM into the fault feature classification of wheel hub drive systems, the RBF kernel function is analyzed to be more suitable for fault classification in this study through two dimensions of training time and testing accuracy.

(3) The method was applied to gear experimental data and achieved good diagnostic results.

(4) The proposed method has been validated through experimental data analysis, but, due to significant differences in vibration characteristics caused by complex working conditions and varying degrees of component damage, further research is needed on the diagnostic effectiveness in actual working environments.

Future work includes fault diagnosis experiments on other components of the wheel drive system, as well as fault diagnosis of wheel drive systems under variable operating conditions in real environments and selecting more multi-dimensional data vectors as input objects for patterns to diagnose wheel drive system faults under imbalanced multi-dimensional data.

**Author Contributions:** Conceptualization, L.X. and W.L.; methodology, L.X., B.Z. and W.L.; software, L.X. and Y.Z.; validation, L.X., Y.Z. and C.L.; formal analysis, C.L.; investigation, L.X. and C.L.; resources, B.Z. and W.L.; data curation, L.X. and W.L.; writing—original draft preparation, L.X. and W.L.; writing—review and editing, L.X., B.Z. and W.L.; funding acquisition, B.Z. All authors have read and agreed to the published version of the manuscript.

**Funding:** This research was funded by the Fundamental Research Funds for the Central Universities, grant number 2019ZDPY08.

**Data Availability Statement:** Data are contained within the article.

**Acknowledgments:** The authors would like to acknowledge the support of Fault Diagnosis and Testing Laboratory in Jiangsu Normal University.

**Conflicts of Interest:** The authors declare no conflict of interest.

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
