# Peer review of "Fault Diagnosis of Mine Truck Hub Drive System Based on LMD Multi-Component Sample Entropy Fusion and LS-SVM"

_actuators, doi:10.3390/act12120468_

Round 1

Reviewer 1 Report

Comments and Suggestions for Authors

In this paper, a fault diagnosis method based on LMD multi-component sample entropy fusion and LS-SVM is proposed. The method can serve as a  reference for the fault diagnosis of rolling bearings and gears inside the wheel drive system. Following comments are provided to authors for improving the quality of the paper. 

(1) I suggest removing Figure 1 and its related description from the introduction section since the main focus of this paper is on the data from the experimental platform.

(2) Please provide a basis for selecting 5 PF components in Section 4.2.

(3) Even though the paper's experimental accuracy can reach 100%, it is crucial to add valuable discussions to convince readers that it can achieve high accuracy in other applications as well. In addition, please also supplement the relevant work directions that the paper can continue to carry out in the future.

(4) The diagnostic accuracy of 100% should not be overemphasized in the third point of the conclusion, as it is based on the experimental results of the paper and has certain limitations.

(5) To attract more attention the paper, authors are encouraged to share data with readers if possible.

Comments on the Quality of English Language

Overall, English expression is relatively fluent. But there is still a need to further polish the language and text of the paper.

Reviewer 2 Report

Comments and Suggestions for Authors

This paper addresses the fault detection algorithm based on the fusion of LMD multi-component sample entropy and LS-SVM, and validates it with experimental data obtained from acceleration responses of the wheel drive system. The literature review is well-written, but there are several controversial issues outlined below:

The theories presented in Chapter 2 provide a simple explanation of key algorithms, but specific parameters or selected criteria are not shown at all. It is recommended to specify them in the application chapter.

In the experimental process, four cases of candidate events are announced, but errors may occur during the replacement of the fault component in the wheel drive system. How can you guarantee the same test conditions? Additionally, the detailed explanation of the test process is not well-prepared.

(Minor) LS-SVM is not explained.

Table 2 shows the main results derived from your proposed methodology, but little discussion is provided. It is important to explain them with a proper physical background. Furthermore, it is recommended to compare them with conventional algorithms, such as LMD decomposition.

How do you determine the precision value in Table 2?

While your highlighted point was the novel fault detection algorithm, the conclusion is still open. Is the accuracy 100%?

My opinion on the current version is negative.

Comments on the Quality of English Language

Minor editing of English language required. 

Round 2

Reviewer 1 Report

Comments and Suggestions for Authors

After the review of the revised manuscript, all my concerns are well addressed. I have no further comments.  

Reviewer 2 Report

Comments and Suggestions for Authors

The raised issues were solved in the revised version. 

Comments on the Quality of English Language

Minor editing of English language required.